# A Method of Modern Standardized Apple Orchard Flowering Monitoring Based on S-YOLO

**Xinzhu Zhou [1], Guoxiang Sun [1,2,]*, Naimin Xu [1], Xiaolei Zhang [1], Jiaqi Cai [1], Yunpeng Yuan [1] and Yinfeng Huang [1]**

1    College of Engineering, Nanjing Agricultural University, Nanjing 210031, China
2    Jiangsu Province Engineering Lab for Modern Facilities of Agriculture Technology & Equipment, Nanjing 210031, China
*    Correspondence: sguoxiang@njau.edu.cn; Tel.: +86-158-5076-0301

**Abstract:** Monitoring fruit tree flowering information in the open world is more crucial than in the research-oriented environment for managing agricultural production to increase yield and quality. This work presents a transformer-based flowering period monitoring approach in an open world in order to better monitor the whole blooming time of modern standardized orchards utilizing IoT technologies. This study takes images of flowering apple trees captured at a distance in the open world as the research object, extends the dataset by introducing the Slicing Aided Hyper Inference (SAHI) algorithm, and establishes an S-YOLO apple flower detection model by substituting the YOLOX backbone network with Swin Transformer-tiny. The experimental results show that S-YOLO outperformed YOLOX-s in the detection accuracy of the four blooming states by 7.94%, 8.05%, 3.49%, and 6.96%. It also outperformed YOLOX-s by 10.00%, 9.10%, 13.10%, and 7.20% for $mAP_{ALL}$, $mAP_S$, $mAP_M$, and $mAP_L$, respectively. By increasing the width and depth of the network model, the accuracy of the larger S-YOLO was 88.18%, 88.95%, 89.50%, and 91.95% for each flowering state and 39.00%, 32.10%, 50.60%, and 64.30% for each type of $mAP$, respectively. The results show that the transformer-based method of monitoring the apple flower growth stage utilized S-YOLO to achieve the apple flower count, percentage analysis, peak flowering time determination, and flowering intensity quantification. The method can be applied to remotely monitor flowering information and estimate flowering intensity in modern standard orchards based on IoT technology, which is important for developing fruit digital production management technology and equipment and guiding orchard production management.

**Keywords:** intelligent agriculture; IoT technology; apple flowering monitoring; open world; swin transformer; SAHI



## 1. Introduction

The quantitative gathering of information on the growing status of fruit trees using current technology facilitates the digital management of orchard production and enhances the precision of orchard production management [1]. Flowering information monitoring is one of the basic techniques for digital orchard management, and it is extensively utilized for orchard flower thinning, pest and disease control, and other management operations. Pruning and intercutting are required to obtain more significant economic returns in the apple-growing industry [2]. In the early stages of apple growth, proper flower and fruitlet thinning may increase the fruit weight per fruit and the blooming yield [3]. Current flower thinning methods mainly include manual thinning [4], chemical thinning [5–7], and mechanical thinning [8].

Traditional flowering monitoring is achieved based on human observations of specific fruit trees at specific times. That is, experts go into the orchard to randomly select a few fruit trees and estimate the flowering state with the eye. After comprehensive consideration,

the overall flowering state of the orchard is obtained. Thinning after 28 days of bloom is ideal for obtaining larger, high-quality Fuji apples [3,9]. However, modern standardized orchards often have large areas and variability in the flowering times of fruit trees in different regions. It is difficult to dynamically adjust orchard flower thinning time and measures for specific fruit tree flowering information, affecting the efficiency and accuracy of modern standardized orchard flower thinning management decisions. Consequently, there is an urgent need for a method that can monitor the various growth stages of apple flowers and quantify the flowering intensity, establishing the groundwork for real-time monitoring utilizing Internet of Things (IoT) technology.

The current apple blossom monitoring methods have not been effectively studied, and most of them have been carried out in simple experimental environments, i.e., with suitable light, shooting angle, and shooting distance, achieving close to 100% detection results. Using closer imaging distances, a study on stamens in fully open flowers [10] disregarded the predictive effect of early buds and semi-open flowers for fully open flowers and found they were only appropriate for close detection. Other studies that have grouped all stages of flowers into one category for detection, even for flower clusters [11,12], have ignored the interaction effects between the flowers at different growth stages and could not accurately monitor the complete flowering process. The division of flowers into three stages of detection [13] ignored the end-flowering stage as a marker of the end of the flowering stage for determining the flowering stage of fruit trees. Other studies have divided apple flowers into 6–8 stages for detection [14,15], devising even more categories. However, the similarity between flowers at different growth stages elevates the cost of data annotation and the inability to count the number after clustering detection in the same category.

Images obtained from closer distances with a high proportion of apple blossom pixels at various stages are simpler to recognize and perform better. However, with the development of IoT monitoring devices, research based on high-resolution images of whole fruit trees acquired at a distance has become an inevitable trend. Current studies have obtained more complete images due to the long imaging distance, such as vehicle-based [16] and uncrewed aerial vehicles [17,18]. However, the tiny area of individual flowers makes the variability between flowers at different growth stages low, and only fully open flowers or even flower clusters are recognized as detection objects. In addition, none of the above studies have observed the entire growth cycle of apple blossoms or tested the models under different weather conditions. The detection models obtained may not be suitable for a wide range of weather conditions. The key to the above problem is that the current detection algorithm cannot effectively detect tiny pixel flowers in high-resolution apple tree images, let alone monitor the complete growth process in complex weather.

Convolutional neural networks are the standard model in computer vision. The related models are categorized into two groups based on whether they directly implement the classification and localization process: the Faster RCNN [19–21] series of two-stage algorithms, the SSD [22], and the YOLO series [23,24] of one-stage algorithms. The current apple flower detection algorithms are primarily separated into mask-based semantic segmentation and box-based object detection. Studies using semantic segmentation algorithms in apple blossom detection have included DeepLab-ResNet [11], Mask R-CNN [12,13], and fully convolutional neural networks [16]. Although these algorithms can segment flowers, they cannot count the number of flowers and are less effective in detecting large aggregations of flowers. The box-based apple flower detection algorithm can count the number of flowers and enable further data analysis. In work using this type of algorithm, the YOLO family, especially YOLO v4 [25], has been widely improved and achieved better detection results [10,15,26,27]. However, these studies have only examined flowers at certain times, lacking the monitoring of the whole flowering process and quantitative analysis of the flowers. Therefore, a method is needed that can accurately detect tiny apple blossoms in high-resolution images and enable multi-stage flower monitoring in the open world.

As the most advanced model in the widely used YOLO family, YOLOX [28] has superior detection performance and has been effectively implemented for similar intensive

detection applications [29,30]. Although CNN models, including YOLOX, have a long history of success in target detection using translation invariance and local correlation, CNN has a restricted field of vision, making it challenging to gather global information. In contrast, Transformer does not have translational invariance and local correlation but can capture long-range dependencies. So Vision Transformer [31] performs better than pure convolutional models for large datasets, especially when massive datasets can be obtained through IoT technology.

Since the introduction of Vision Transformer, many works have tried combining CNN and Transformer to motivate the network to inherit the advantages of CNN and Transformer and retain the global and local features to the maximum extent. As a landmark work, Swin Transformer [32], with shifted windows as a prominent feature, was created. With self-attention at its foundation, Swin Transformer gathers global contextual information to establish long-range dependency on targets and extract more robust features, demonstrating the potential to replace traditional convolutional networks as the new backbone network in computer vision.

In order to achieve information monitoring of the complete flowering process using IoT technology, research based on high-resolution images of apple trees taken in complex weather is essential. However, such images are not only tough to obtain, but also the typical characteristics, such as a complex and changeable environment, a tiny proportion of flower pixels, and hazy texture and color detail information, provide obstacles for flower detection and monitoring. This study took high-resolution images of apple blossoms at the complete growth stage in the open world as the research object and used the Slicing Aided Hyper Inference (SAHI) algorithm to generate mixed datasets containing global and local information. Then an S-YOLO model was designed based on Swin Transformer, achieving the accurate detection of apple blossoms at four growth stages. An analysis model for the number and number share of apple blossoms at each stage was established, realizing the flowering intensity and flowering monitoring of the orchard or even specific fruit trees. This work gives further theoretical and technological support for monitoring orchard flowering growth using IoT technology.

## 2. Materials and Methods

### 2.1. Experimental Design

As shown in Figure 1, the specific workflow of the apple flowering stage monitoring method is as follows: Step 1: obtain images of complete apple trees in the experimental area of the orchard at the flowering stage. Step 2: label the obtained images according to bud, half-open, fully open, and end-open using labelimg, and obtain the labeled image file. Step 3: slice the annotated image using the SAHI algorithm and blend it with the original image. Step 4: build the S-YOLO model and use the blended dataset for model training and validation. Step 5: use the trained model for apple blossom detection. Step 6: perform data analysis on the detection results. Step 7: implement flowering intensity estimation and flowering period monitoring.

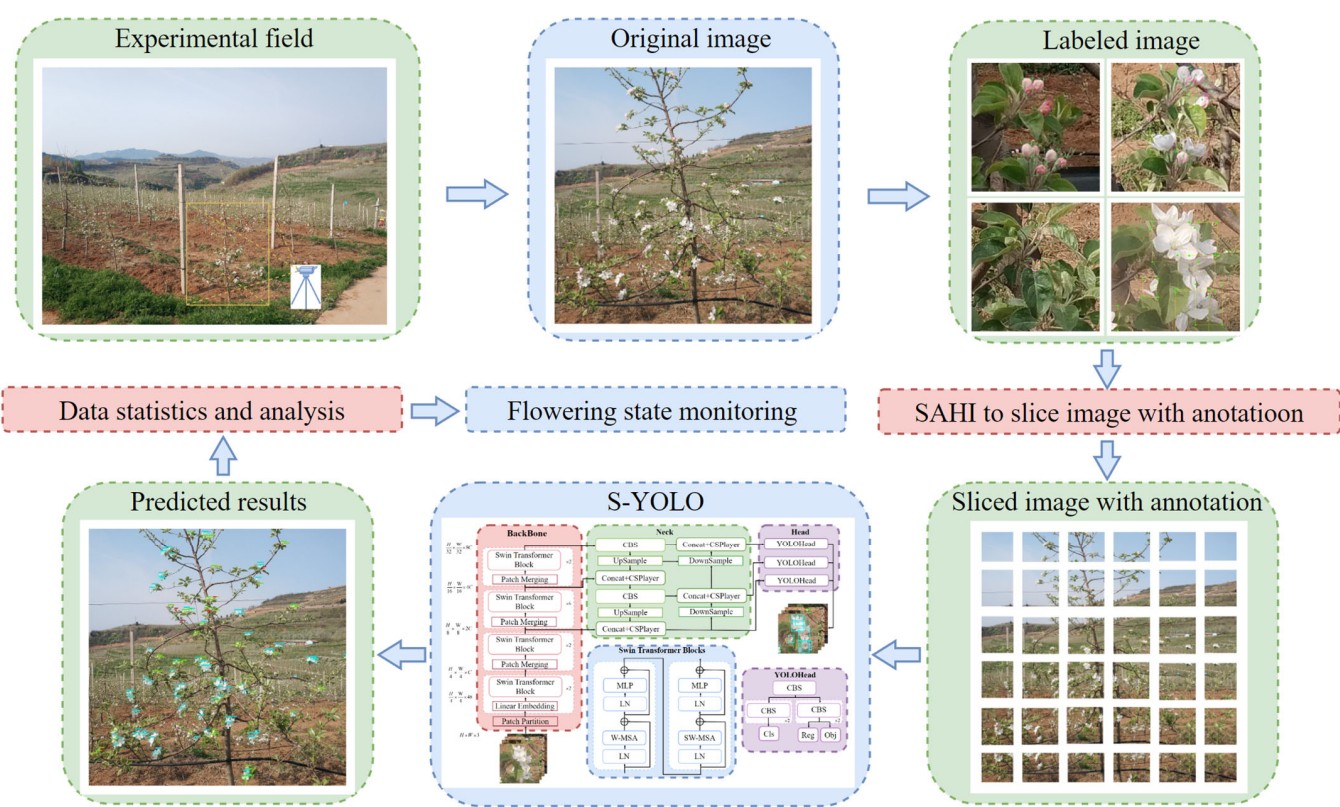

**Figure 1.** Workflow for monitoring the flowering state of modern standardized apple orchards.

*2.2. Image Data Acquisition*

The images utilized in this research were captured between 3 April 2020 and 16 April 2020, in the First Asian Alpine Orchard in Shihe Township, Lingbao City, Henan Province (E 111°4′2.6976″, N 34°27′1.44″), while the whole orchard was in the first flowering stage. The subjects were 115 red Fuji apples from 5-year-old trees in rows 4.2 m apart with 2.1 m between the plants. To simulate IoT devices to acquire images of apple trees, the investigator stood 2.5 to 3 m away from the tree's roots and used a mobile phone to capture photographs of the blooming stage of the apple trees.

The camera was operated from 10:00 am to 12:00 pm on the 13 days of shooting, and 115 images were obtained daily. A total of 1494 images of 3000 × 3000 pixels of apple trees under different weather conditions, including sunny, cloudy, and rainy days, were obtained in this study. Table 1 presents the precise weather information.

**Table 1.** Weather on various dates of shooting.

| Date | Weather | Light Intensity | Temperature | Windy |
|------|---------|-----------------|-------------|-------|
| 0403 | Overcast | Weak | 8–16° | No |
| 0404 | Sunny | Strong | 8–18° | Yes |
| 0405 | Sunny | Strong | 9–20° | Yes |
| 0406 | Sunny | Strong | 9–20° | Yes |
| 0407 | Sunny | Strong | 9–20° | Yes |
| 0408 | Sunny | Hazy | 11–25° | No |
| 0409 | Sunny | Normal | 8–24° | Yes |
| 0410 | Light rain | Weak | 5–10° | Yes |
| 0411 | Light rain | Weak | 5–15° | Yes |
| 0412 | Sunny | Strong | 5–20° | Yes |
| 0413 | Sunny | Strong | 9–23° | No |
| 0415 | Sunny | Normal | 12–25° | No |

The flowering status of apple trees is divided into four stages: the first flowering stage, the middle flowering stage, the full flowering stage, and the last flowering stage, which correspond to the four growth statuses of apple flowers. The first stage of the apple flower is the bud (Figure 2a, red arrow), and they become half-open at the second stage (Figure 2a, green arrow and Figure 2b, green arrow) when the buds swell into white or light pink blooms that look like balloons. Once the petals have unfurled in the bud, the flower enters the fully open stage (Figure 2c, blue arrow), and the end-open stage is when the petals drop completely (Figure 2d, purple arrow). This experiment labeled 39,980 instances of the four flowering stages on 317 3000 × 3000 pixels images of Fuji apple trees using labelimg (Table 2), fulfilling the criterion that at least 3000 to 4000 instances of each class must be labeled in complicated agricultural contexts, as suggested in [27].

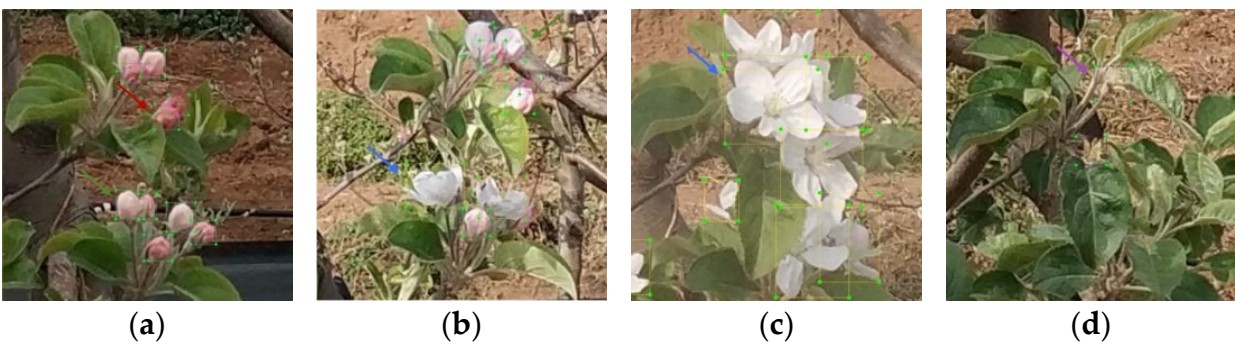

**(a)**        **(b)**        **(c)**        **(d)**

**Figure 2.** Flower images of apple tree at four flowering stages (300 × 300 pixels). (**a**) First flowering stage; (**b**) middle flowering stage; (**c**) full flowering stage; (**d**) last flowering stage.

**Table 2.** Unprocessed annotation data information.

| Class | Bud | Half-Open | Fully Open | End-Open |
|---|---|---|---|---|
| Number | 11,865 | 10,885 | 12,288 | 4942 |
| Aspect ratio (pixels) | 17.08:17.23 | 22.32:22.30 | 49.36:48.61 | 27.68:27.13 |
| Average area (pixels$^2$) | 318.86 | 541.71 | 2586.45 | 798.06 |
| Area ratio (%) | 0.0035 | 0.0060 | 0.0287 | 0.0088 |

Table 2 displays the raw annotation category information, including the number of annotations, the aspect ratio of various flowers, and the ratio of the area occupied in the image. The number of flower annotations in the first three growth stages was close, except for the last bloom. The aspect ratio is the ratio of the labeled flower length to the width, with the same category of flowers possessing a length-to-width pixel ratio near 1:1 and an average pixel size ratio among the different flower stages of 17:22:49:27. Therefore, the order of the size of the average area is fully open > half-open > end-open > bud. The fully open stage has the highest number of annotations and largest average area. However, its ratio in the image was less than 0.03 %, and the other three flowering stages made up less than 0.01 %, making accurate flower detection more challenging.

From the overall view of the annotation information, the image data utilized in the experiment and the annotated data were of high quality, which provided practical support for the training and validation of the model. This dataset has the following characteristics:

- The dataset used for the experiment covered a range of weather conditions, the apple tree's growth postures, and the complete flowering process.
- The original image was a high-resolution image of 3000 × 3000 pixels, where the vast majority of flowers were almost always smaller than 50 × 50 pixels.
- The flowers at each growth stage were manually labeled with sufficient numbers and fineness. All factors affecting apple flower detection, such as biometric features, gestures, shadows, and light, were considered at the human level.

### 2.3. Slicing Using SAHI

The high-resolution apple tree images used in this study contained a large and dense number of tiny pixel flowers. The higher resolution made directly inputting the photos into the network to extract features too computationally costly. However, reducing the resolution would result in losing information on details related to the flowers.

Multiple solutions have been developed to address the problem of small, dense objects in high-resolution photographs. The traditional method of filling and then segmenting images [6] and the method of copying and enhancing [33] images after oversampling require segmenting a large number of annotations, which results in a large number of features being altered to the point of being incompatible with the original dataset. Enlarging the target region [34] can enrich small object features, but it will add additional computational volume and is challenging to adapt to the demand for detection speed in some agricultural fields. To preserve image detail information and reduce the model calculation costs, the SAHI slicing algorithm was used to increase the model detection accuracy.

The SAHI [35] algorithm is a slicing-assisted inference approach for object detection models that perform inference by cropping the images and performing inference on them. The most notable benefit of SAHI is that it can be used in any object detection inference method, considerably enhancing the detection level of small targets while just linearly lengthening the computation time in slices. The SAHI algorithm effectively increases the precision of YOLO series detection [36].

Considering the efficacy of SAHI in small object detection applications, the SAHI algorithm was applied with a 20% coverage to the dataset utilized in this study, yielding 640 × 640 pixel images (Figure 3).

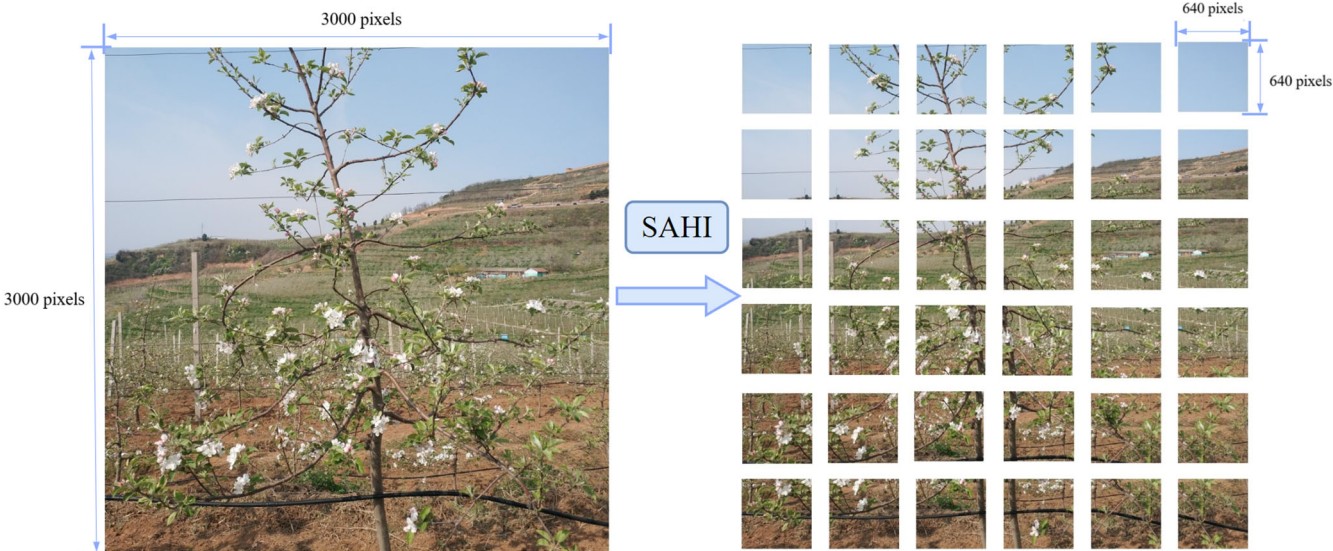

**Figure 3.** The process of SAHI splitting the original image.

The SAHI algorithm divides the original image of 3000 × 3000 pixels into 640 × 640 pixels, which can be directly fed into the network, eliminating the computational overhead of huge images without scaling, and minimizing the loss of detail information due to resizing by approximately five times. The dataset created by combining the original and sliced images is guaranteed to contain large images (3000 × 3000) with high semantic information and small images (640 × 640) with detailed local information. Additionally, there is a 20% overlap area between the sliced images which can also facilitate information fusion.

### 2.4. S-YOLO Detection Model

#### 2.4.1. Model Construction

Swin Transformer can better capture global semantic information than traditional convolutional neural networks and can better fuse global and local information and extract

more powerful features. To reduce the number of model parameters to almost the same level as that of the pure convolutional backbone network to ensure the fairness of the experimental results, Swin Transformer-tiny was substituted for the YOLOX backbone network to generate the S-YOLO model. The overall architecture of the S-YOLO is shown in Figure 4.

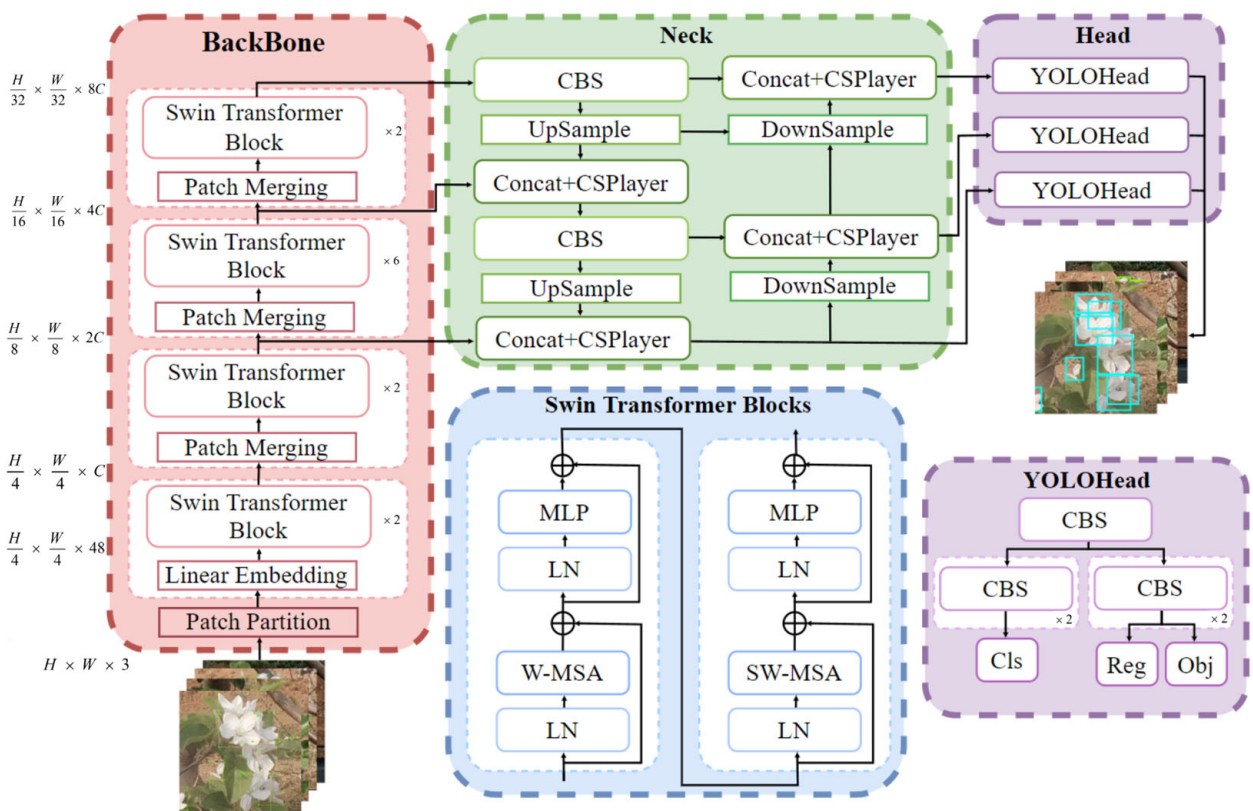

**Figure 4.** S-YOLO network structure.

S-YOLO is separated into four distinct size models based on the varying depths and widths of the neck and head channels. The corresponding depth and width ratios for different versions of the model are S-YOLO-tiny (S-YOLO-t: 0.33, 0.375), S-YOLO-small (S-YOLO-s: 0.33, 0.50), S-YOLO-middle (S-YOLO-m: 1.00, 1.00), and S-YOLO-large (S-YOLO-l: 1.33, 1.20).

The backbone network consists of five parts. The first part is the patch partition, and the last four parts are composed of two consecutive Swin Transformer blocks, where patch partition and linear embedding are equal to Patch Merging, which plays the role of downsampling. Swin Transformer substituted the conventional multi-head self-attention (MSA) module with the window-based multi-head self-attention (W-MSA) module and the shifted window-based multi-head self-attention (SW-MSA) module. Each Swin Transformer block consists of three components linked by a residual structure: a LayerNorm (LN) layer, a W-MSA/SW-MSA module, and a multilayer perceptron (MLP) with two completely connected layers and GELU nonlinearity. Using alternating transformations of W-MSA and SW-MSA to conduct all attention actions in a given window, the Swin Transformer block minimizes the computing volume of the model.

Detail information, such as color and texture, is crucial for flower detection. Therefore, retaining and extracting as much shallow information as possible becomes a vital consideration while building a feature extraction network. The neck part used a PAnet structure [37] to accomplish a twofold sampling of the features and to improve the network's capacity to fuse features. The CBS module, which comprises a convolutional layer (Conv), a batch normalization layer (BN), and an activation function called SiLU, is the

primary module for feature extraction in the S-YOLO neck and head sections. The CSPlayer layer, using the idea of residuals, consists of two parallel CBS modules and multiple residual units in a series, which will play a role in PAnet for better image features and fusion capability extraction.

The design of the detecting head module is centered on efficiently employing the characteristics gathered from global and local information. The YOLOHead part employs decoupled heads with quicker convergence and greater precision. The decoupled head is controlled via CBS modules with varying channel counts and partitioned into classification and regression subnetworks. The classification subnetwork calculates the probability of detecting flowers belonging to distinct classes (Cls) of flower labels. In contrast, the regression subnetwork predicts the feature points' classes (Obj) and positions (Reg). Combining three sets of YOLOheads designed to detect flowers of various sizes produced a considerable number of suggestion boxes, and the anchor-free SimOTA algorithm provided the final detection results.

The hybrid dataset generated using SAHI was fed into the S-YOLO network for training and validation after data enhancement via the Mosaic [25] and MixUp [38] algorithms.

### 2.4.2. Model Training and Validation Environment

The mixed dataset was divided into a training set, a validation set, and a test set, at a ratio of 64:16:20. The training and validation sets were resized to $640 \times 640$ pixels for input to the network for training and validation. The test set was used for the network assessment and testing. The model training process was carried out using the Ubuntu 18.04 Cloud operating system (Cuda 11.0, Cudnn 8.0.4, Python 3.8, Pytorch 1.8, $4 \times$ NVIDIA RTX 3090). The model assessment process was performed using the Windows 11 operating system (Cuda 11.3, Cudnn 8.2.1, Python 3.8, Pytorch 1.10.2, $1 \times$ NVIDIA RTX 1650). During the experiments, the freezing and unfreezing training process was conducted with 100:200 epochs, with primary batch sizes of 128:32 and 16:16 when the SAHI algorithm was not utilized. The other critical hyperparameters are listed in Table 3.

**Table 3.** The primary hyperparameters of the model training process.

| Hyperparameter | Value |
|---|---|
| Initial learning rate | 0.01 |
| Minimum learning rate | 0.0001 |
| Optimizer | sgd |
| Momentum | 0.937 |
| Weight decay | 0.0005 |
| Learn rate decay type | cos |

### 2.4.3. Evaluation Indicators

Precision (P), recall (R), average precision (AP), and different types of mean average precision (*mAP*) are the main indicators used to assess the efficacy of the model for detecting apple flowers. The experimental outcome measures can be presented by calculating various combinations of positions within the confusion matrix: true positive (TP) is the accurate forecast for positive samples, true negative (TN) is the correct prediction for negative samples, false positive (FP) is the erroneous prognosis for positive samples, and false negative (FN) is the incorrect prediction for negative samples. In addition, the precision–recall (P–R) curve forms corresponding points between the horizontal axis, representing recall (R), and the vertical axis, representing precision (P) (with an IoU threshold equal to 0.5). Different intersections over the union (IoU) values were obtained by setting a certain degree of overlap between the prediction and the ground truth. Other specific metrics were calculated as follows:

AP: Average precision for a single category (IoU threshold from 0.5 to 0.95 in steps of 0.05), including bud, half-open, fully open, and end-open apple flowers;

$mAP_{\text{ALL}}$: Mean average precision of apple flowers of the four stages (all pixels);

$mAP_S$: $mAP$ for small objects whose area is smaller than 322 pixels;
$mAP_M$: $mAP$ for medium objects whose area is between 322 pixels and 962 pixels;
$mAP_L$: $mAP$ for large objects whose area is bigger than 962 pixels.

Calculating the *P*, *R*, *AP*, and mAP for a given IoU threshold is defined in Equations (1)–(4).

$$P = \frac{TP}{TP + FP} \tag{1}$$

$$R = \frac{TP}{TP + FN} \tag{2}$$

$$AP = \sum_{k=1}^{N} \max_{\widetilde{k} \geq k} \left[ P(\widetilde{k}), P(k) \right] \left[ R(\widetilde{k}) - R(k) \right] \tag{3}$$

$$mAP = \frac{\sum_{i=1}^{M} AP_i}{M} \tag{4}$$

where $k$ and $\widetilde{k}$ represent the point serial numbers before and after interpolation; $N$ is the number of images under a category; $M$ is the number of categories; $i$ is the category label; and $P(k)$ and $R(k)$ are the precision and recall of the kth point. $AP_i$ is the average precision of class $i$.

### 2.5. Apple Flowering Monitoring

In this experiment, the trained model was applied to 1494 images of apple trees taken at different times to detect flowers at different stages. The prediction of each apple tree image using S-YOLO-s to obtain the boxes corresponding to the four stages of flowers in each image and accumulating the number of boxes in the same category in all images on the same date obtained the total number of flowers in the four stages. On this basis, the total number of apple blossoms at each stage was divided by the number of images to obtain the average number of flowers at each stage in each image.

It is possible to determine the relative number share by evaluating the proportional relationships between the various stages of flowers within a single image. When the percentage of flowers at a particular stage in an image surpasses fifty percent, the fruit tree is deemed at the corresponding flowering stage. The day with the highest number of apple blossoms at a particular stage of the growth cycle is the peak time for apple blossoms at that stage. Among all images of fruit trees under a specific date, the average number of flowers in the four stages can be used to determine the flower proportion, and thus, the overall flowering status of the orchard. The percentages of fully open flowers correspond to the flowering intensities from 0 to 100, and this precise quantitative index will provide data support for flower-thinning decisions.

## 3. Results and Discussion

### 3.1. Image Slice Results

The flower images were sliced using the SAHI algorithm and combined with the original images to create a hybrid dataset (Table 4). The relevant feature changes were recorded before and after the slicing (Table 5).

**Table 4.** Annotated data information after slicing using the SAHI algorithm.

| Class | Bud | Half-Open | Fully Open | End-Open |
|---|---|---|---|---|
| Number | 32,060 | 29,402 | 34,703 | 13,648 |
| Aspect ratio (pixels) | 17.55:17.70 | 22.64:22.68 | 48.51:47.85 | 27.97:27.46 |
| Average area (pixels$^2$) | 334.50 | 557.75 | 2485.52 | 813.22 |
| Area ratio (%) | 0.0542 | 0.0894 | 0.3934 | 0.1312 |

**Table 5.** Changes in annotated data information after SAHI algorithm slicing.

| Class | Bud (%) | Half-Open (%) | Fully Open (%) | End-Open (%) |
|---|---|---|---|---|
| Number | +170.21 | +170.11 | +182.41 | +176.16 |
| Aspect ratio | +2.76: +2.73 | +1.43: +1.70 | −1.72: −1.56 | +1.05: +1.27 |
| Average area | +4.90 | +2.96 | −3.90 | +1.90 |
| Area ratio | +1428.91 | +1385.91 | +1268.89 | +1379.58 |

Following the SAHI algorithm slice, the percentages of flowers in each category grew by 170.21%, 170.11%, 182.41%, and 176.16%, respectively, resulting in 109,813 high-quality labeled data (a 150% increase) to the network. While allowing for a higher batch size for training, the changes in the aspect ratio and average area were within 3.00% and 5.00%, and the corresponding change in the area ratio was at least 1268.89%. In addition, the fully open stage had the largest average single flower area, which was 7.43 times bigger than the bud stage, 4.46 times bigger than the half-open stage, and 3.06 times bigger than the end-open stage. The high area of full blooms prompted the SAHI algorithm to split the fully open flowers that were at the boundary of the cut area into multiple parts and to consider them as newly fully open. This split came at the cost of a 3.90% reduction in the average areas, prompting the most significant increase in the number of fully open flowers. Fully open was the only flower growth stage with a negative average area increase.

### 3.2. Flower Detection with S-YOLO

3.2.1. Comparison with YOLOX-s

In the COCO dataset, the superiority of YOLOX over sophisticated models, such as PPYOLO [39], YOLOv3, and EfficientDet [40], was established [28]. YOLOv4 was proven to have greater precision and *mAP* than Faster R-CNN and SSD 300 for detecting apple pistils [10]. Consequently, the comparison experiment portion of this study was performed on the original YOLOX-s model and the modified S-YOLO-s, and the pertinent data were collected (Table 6).

**Table 6.** Comparison of the effects of YOLOX-s and S-YOLO-s on flowering detection in apples.

| Model | P-Bud [1] (%) | P-Half-Open (%) | P-Fully Open (%) | P-End-Open (%) | $mAP_{ALL}$ (%) | $mAP_{S}$ (%) | $mAP_{M}$ (%) | $mAP_{L}$ (%) |
|---|---|---|---|---|---|---|---|---|
| YOLOX-s [2] | 79.19 | 81.27 | 85.21 | 83.90 | 27.40 | 21.40 | 36.00 | 58.90 |
| S-YOLO-s | 87.13 | 89.32 | 88.70 | 90.86 | 37.40 | 30.50 | 49.10 | 66.10 |

[1] Represents the precision of the bud, the same as below. [2] Note: The SAHI algorithm was used in S-YOLO by default and YOLOX by contrast.

Figure 5a shows the loss curves during the training of the S-YOLO-s model. After the 100th epoch, the backbone started to thaw, and the losses of the training set and the validation set fell. The validation set loss was higher than the training set after the 170th epoch, and when the decreasing trend was smaller than the training set, the model started to overfit. Figure 5b shows the P–R curves with IoU = 3. The AP of the fully open flowers was significantly higher than the other three stages, and the differences among the AP values of the three stages were not significant. This phenomenon was significantly correlated with the higher pixel proportion of the fully open flowers.

The precision of S-YOLO-s was enhanced by 7.94%, 8.05%, 3.49%, and 6.96%, and different types of *mAP* by 10.00%, 9.10%, 13.10%, and 7.20%, respectively, compared to YOLOX-s at each flowering stage. By comparing the experimental results, it was found that the improved model's apple flower detection precision was significantly higher than the original model and indirectly higher than EfficientDet, Faster R-CNN, and SSD 300. Therefore, it is practical and feasible for the model to accurately detect apple flowers with high resolution. The detection results of YOLOX-s and S-YOLO-s for the apple tree blossoming images under four typical weather conditions of overcast, sunnier, foggy, and sunny days (Figure 6a,b) provide further proof of the model's superiority.

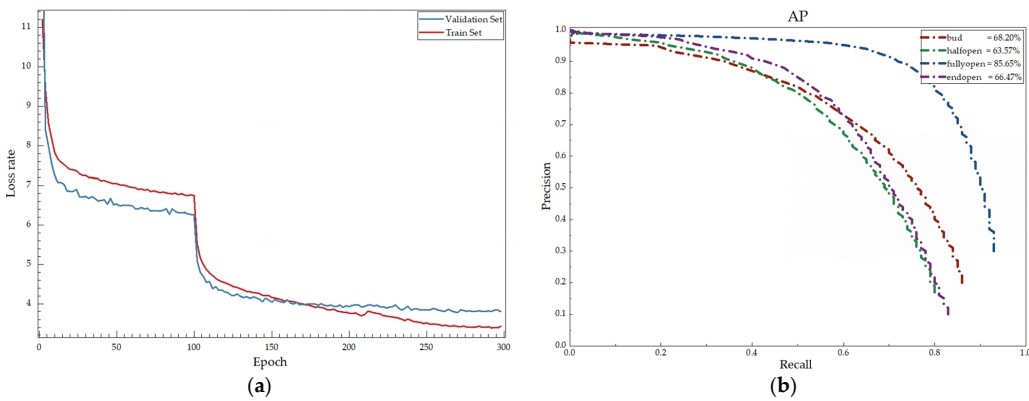

(**a**)　　　　　　　　　　　　　　　　(**b**)

**Figure 5.** Loss and P–R curve. (**a**) Loss curve of S-YOLO-s. (**b**) P–R curves with mixed datasets (IoU = 0.3).

(**a**)

(**b**)

(**c**)

(**d**)

**Figure 6.** Detection of four growth stages of apple blossoms with red, green, blue, and purple boxes using different models. (**a**) YOLOX-s detection results; (**b**) S-YOLO-s detection results; (**c**) YOLOX-l (add SAHI) detection results; (**d**) S-YOLO-l detection results.

### 3.2.2. The Results of Different Versions of Models

The results of the comparison experiments demonstrate that the S-YOLO model performed better in detecting apple flowers at various growth stages. An ablation experiment was designed to gain a deeper understanding of how the two improvements of adding the slicing algorithm and replacing the backbone network contributed to the results and to determine how both improvements could be used more effectively to produce better results.

Table 7 shows the results of the ablation experiments. Using YOLO-s as the baseline, the SAHI algorithm improved the precision by 0.70%, 0.04%, 1.04%, and 3.38% for each flowering stage and by 6.70%, 5.50%, 7.20%, and 7.70% for the different types of *mAP*, respectively. The results indicate that the SAHI algorithm can successfully enhance the detection impact of the model, and the degree of enhancement was proportional to the object's pixel size. After replacing the original backbone network with Swin Transformer-tiny, the model parameters took up 35.79 MB, the FLOPs took up 95.57 GB, the precision improved by 7.24%, 8.01%, 2.45%, and 3.58%, and the *mAP* improved by 3.30%, 3.60%, 5.90%, and −0.50%. After rebuilding the backbone, the results indicate that S-YOLO was more sensitive to detecting small objects of varying length and breadth. The promotion of detection precision by using Swin Transformer as a backbone network was negatively correlated with the object size, and the enhancement of the *mAP* showed an inverted U-shape with flower size, which led to negative growth of the $mAP_{\text{L}}$.

**Table 7.** Results of ablation experiments.

| Model | P-Bud (%) | P-Half-Open (%) | P-Fully Open (%) | P-End-Open (%) | $mAP_{\text{ALL}}$ (%) | *mAP*s (%) | *mAP*m (%) | *mAP*l (%) | Parameters (M) | FLOPs (G) |
|---|---|---|---|---|---|---|---|---|---|---|
| YOLOX-s | 79.19 | 81.27 | 85.21 | 83.90 | 27.40 | 21.40 | 36.00 | 58.90 | 8.94 | 26.64 |
| YOLOX-s (+SAHI) | 79.89 | 81.31 | 86.25 | 87.28 | 34.10 | 26.90 | 43.20 | **66.60** | 8.94 | 26.64 |
| S-YOLO-s | 87.13 | **89.32** | 88.70 | 90.86 | 37.40 | 30.50 | 49.10 | 66.10 | 35.79 | 95.57 |
| S-YOLO-t | 79.38 | 80.50 | 83.02 | 84.02 | 32.40 | 25.60 | 41.50 | 57.10 | 30.80 | 80.58 |
| S-YOLO-m | 81.95 | 83.48 | 86.96 | 86.86 | 35.10 | 28.20 | 45.40 | 65.70 | 45.89 | 135.35 |
| S-YOLO-l | **88.18** | 88.59 | **89.50** | **91.95** | **39.00** | **32.10** | **50.60** | 64.30 | 51.37 | 157.68 |
| Swin-S | 82.35 | 84.02 | 85.60 | 87.37 | 34.50 | 27.60 | 44.10 | 65.70 | 57.07 | 165.84 |

While replacing various backbones and detection heads extended the model, the detection effect exhibited a non-linear correlation with the model size. Different sizes of S-YOLO were obtained by adjusting the channel depth and width variation of YOLOX while maintaining increased data using SAHI. The *mAP* values from S-YOLO-t to S-YOLO-l were 32.40%, 37.40%, 35.10%, and 39.00%. This phenomenon, where the *mAP* did not grow as the model grew, resulted from the uncoordinated channel change between the network's backbone and neck. Swin-S replaced the YOLOX-s backbone with Swin Transformer-small. Although the number of parameters and FLOPs of the Swin-S model were higher than S-YOLO-l, the precision and *mAP* were smaller than S-YOLO-s. Therefore, the appropriate ratio of the number of structural parameters and the channel variation are necessary factors for the S-YOLO variant to achieve higher detection results.

In summary, S-YOLO-s outperformed the original model in detecting each flower stage at a high resolution, which resulted from the combined effect of the SAHI algorithm increasing the percentage of flower pixels while keeping the image features unchanged and the Swin Transformer being used as the backbone network. The high-resolution local information provided by SAHI without scaling was fed to the network along with the global information of the scaled original image. This information was fused via the Swin Transformer and subsequently fully used by the network, prompting the model to produce state-of-the-art experimental results. Moreover, S-YOLO is exceptionally sensitive to the size of the detected object, and larger detected objects will get a minor boost or even negative growth in the *mAP* compared to other objects of smaller length and width. Ablation experiments revealed the superiority of the S-YOLO performance and illustrated that appropriate channel depth variation and balanced parameter scaling will provide better results than arbitrarily expanding the model. This experiment provides data to

support the replacement of the larger Swin Transformer as the backbone to obtain greater experimental results.

### 3.2.3. Comparing the Effects of Other Measures

The detection results on the mixed dataset (Table 8) show that the improved S-YOLO-l was slightly less effective than YOLOX-l with the comparable sizes of parameters and FLOPs on the mixed dataset, but this does not mean that the improvement of the model was unsuccessful. The dataset used for model evaluation in this experiment consisted of a $640 \times 640$ pixel image after slicing and the original $3000 \times 3000$ pixel image. However, the actual detection object in the natural environment should be $3000 \times 3000$ pixels, so the data in the test dataset were replaced with the raw dataset and the experiment was conducted again to obtain the new results.

**Table 8.** Comparison results of YOLOX-l and S-YOLO-l with similar parameters (54.15 M and 51.37 M) on mixed and raw datasets (all using SAHI).

| Model | Datasets | P-Bud (%) | P-Half-Open (%) | P-Fully Open (%) | P-End-Open (%) | $mAP_{ALL}$ (%) | $mAP_S$ (%) | $mAP_M$ (%) | $mAP_L$ (%) |
|---|---|---|---|---|---|---|---|---|---|
| S-YOLO-s | Mixed | 87.13 | **89.32** | 88.70 | 90.86 | 37.40 | 30.50 | 49.10 | 66.10 |
| YOLOX-l | Mixed | 85.27 | 88.66 | **92.02** | **92.92** | **41.70** | **35.60** | **55.40** | **74.60** |
| S-YOLO-l | Mixed | **88.18** | 88.59 | 89.50 | 91.95 | 39.00 | 32.10 | 50.60 | 64.30 |
| YOLOX-l | Raw$_{20\%}$ * | 81.00 | 85.80 | 90.30 | 90.00 | 34.30 | 28.30 | 48.20 | 71.80 |
| S-YOLO-l | Raw$_{20\%}$ | 84.30 | 87.78 | 90.56 | 91.88 | 34.40 | 28.40 | 48.20 | 70.60 |

* The proportion of the test set used for all datasets was 20%.

The results in Table 8 show that S-YOLO-l significantly outperformed YOLOX-l in precision and achieved better results in all types of *mAP* after adjusting the test set percentage to 20%. Therefore, the improved model still outperformed the original model in the high-resolution task of detecting the growth stage of apple blossoms, even when the gain from the SAHI algorithm was ignored. In addition, if the SAHI algorithm is not used, YOLOX-l will not be trained properly due to the low number of input images. Notably, the backbone network utilized in this investigation was Swin Transformer-tiny. With enough computing resources, labeled data, and disregarding the negative impact of a bigger model, a bigger S-YOLO network with the proper parameter scaling and number of channels would produce superior detection results.

Figure 6c,d shows the detection results of YOLOX-l (add SAHI) and S-YOLO-l for the apple tree flowering images under four typical weather conditions: cloudy, more sunny, foggy, and sunny days. The image presentation results indicate that YOLOX-l, with the inclusion of the SAHI algorithm, may have performed marginally better than S-YOLOX-l at several detection locations. Notably, the initial annotation volume utilized in the experiment was 39,980, which grew to 109,813 after slicing using the SAHI technique and mixing with the original dataset, but was significantly less than the COCO dataset. Therefore, S-YOLO-l cannot fully exploit its strengths. With the development of IoT technologies, the issue of a too-small training dataset will be resolved, and it will also be possible to use bigger Swin Transformer models to obtain better detection results. By observing Table 8 and Figure 6, it is possible to infer that the enhanced S-YOLO-l beat YOLOX-l in an identical situation using the SAHI method but that the existing quantity of data did not cause S-YOLO to demonstrate an overwhelming advantage.

### 3.3. Apple Flowering Monitoring Results

A total of 1494 fruit tree images ($3000 \times 3000$ pixels) were fed into S-YOLO-s for flower prediction to obtain the number variation and proportional variation under a time series (Figure 7b). Except for the decline in the number of buds caused by the late shooting date, the number of flowers in all three categories showed arching characteristics in line with the natural pattern for time variation. The average flower density at the peak of each stage was

55.687 on 3 April, 47.565 on 6 April, 118.183 on 9 April, and 17.522 flowers/tree on 15 April (Figure 7a) corresponding to 75.7%, 46.7%, 82.26%, and 49.58% of all flowers, respectively (Figure 7b). The flowering intensities of the orchard on different dates were 1.13%, 3.94%, 19.54%, 37.34%, 57.57%, 72.18%, 82.26%, 81.59%, 75.74%, 77.04%, 76.59%, 47.83%, and 12.18%. The experimental results show that the orchard was in the first flowering stage on 3 April, the middle flowering stage on 5 April, the full flowering stage on 7 April, and the last flowering stage on 15 April. In addition, based on the trend of the number of end-open flowers, it was predicted that only these would remain in the orchard from 17 to 18 April, which means that the orchard would be in the last flowering stage completely.

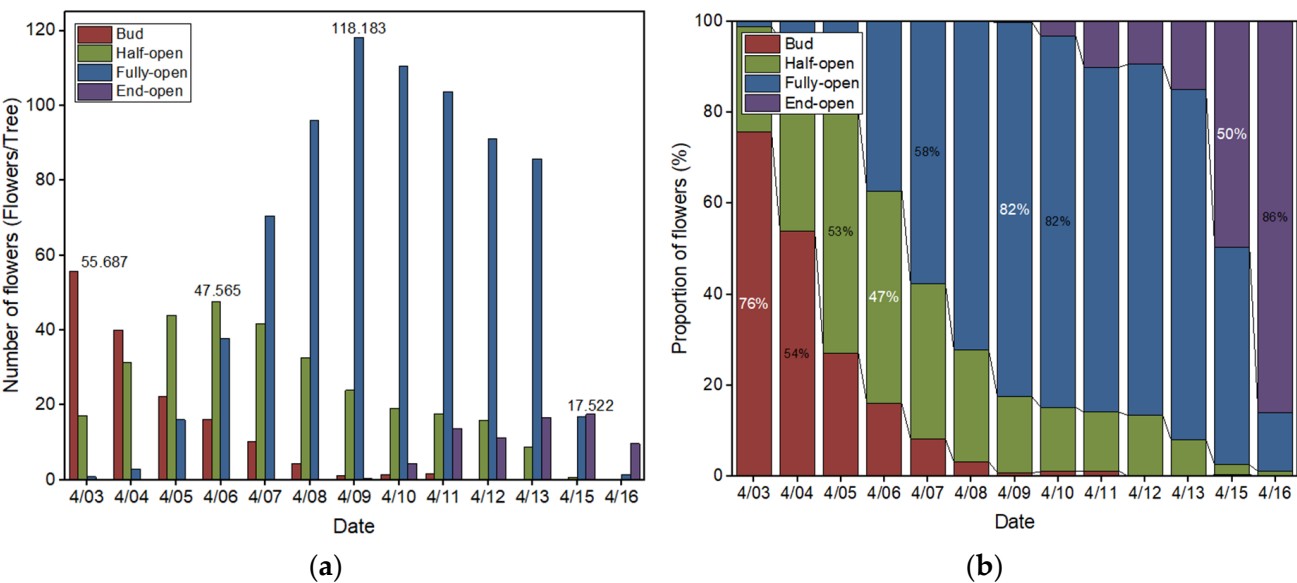

**Figure 7.** Flowering stage changes. (**a**) Changes in the daily number of flowers at each stage; (**b**) changes in the daily percentage of flowers at each stage.

This experiment demonstrated that with the help of an S-YOLO-s high-performance detector, it was possible to obtain time-series changes in the number and proportion of flowers. At the same time, it was possible to achieve the daily flowering intensity estimation and flower peak time determination at each stage, and finally, flowering information monitoring was realized. Notably, the results mentioned above were achieved by counting the detection results of single fruit trees; thus, S-YOLO-s is adequate for identifying the blooming phases of single fruit trees.

## 4. Conclusions

This study proposes a Transformer-based apple flowering monitoring method for monitoring the whole flower growth process of full fruit trees in the open world. In this work, the non-pure convolutional S-YOLO was model used to detect the four growth stages of apple blossoms accurately and to analyze the changes in the numbers and percentages of blossoms at each growth stage in order to estimate the peak flowering time and flowering intensity and to complete the monitoring process. The main conclusions are as follows.

1.  Based on the combination of YOLOX and Swin Transformer, the SAHI algorithm was added to form the S-YOLO model. S-YOLO-s improved the precision compared to the original YOLOX-s by 7.94%, 8.05%, 3.49%, and 6.96% for the four flowering states and by 10.00%, 9.10%, 13.10%, and 7.20% for the $mAP_{ALL}$, $mAP_S$, $mAP_M$, and $mAP_L$, respectively. S-YOLO-l resulted in 88.18%, 88.95%, 89.50%, and 91.95% precision at each flowering state and 39.00%, 32.10%, 50.60%, and 64.30% for each type of $mAP$, respectively. Without considering the SAHI algorithm boost, the non-pure convolutional S-YOLO-l model slightly outperformed the YOLOX-l model with similar parameters and FLOPs in the original dataset, with improvements of 3.30%, 1.98%,

0.26%, and 1.88% in detection precision. In addition, using a bigger Swin Transformer as the backbone, designing an appropriate percentage of structural parameters, and collecting more training data may have resulted in improved experimental outcomes.

2. The SAHI algorithm made the object-detected aspect ratio and average area vary between 3.00% and 5.00%, respectively, while increasing the image area ratio by 1250%. The SAHI algorithm increased the number of annotations of flowers in the four growth stages by 170.20%, 170.11%, 182.41%, and 176.16%, respectively, and the total amount of annotated data increased by 150% to 109,813, providing more quality data for the model training process. The experimental results show that the SAHI algorithm improved the precision by 0.70%, 0.04%, 1.04%, 3.38%, and the *mAP* by 6.70%, 5.50%, 7.20%, 7.70% for each flowering stage, respectively, and the larger the object detected, the more the detection effect was improved.

3. Using the results of S-YOLO, the quantity and percentage of apple flowers and the flowering intensity were estimated daily for each stage of the orchard during the flowering period, and the peak time was identified. The average flower density at the peak of each stage was 55.687 on 3 April, 47.565 on 6 April, 118.183 on 9 April, and 17.522 flowers/tree on 15 April, corresponding to 75.7%, 46.7%, 82.26% and 49.58% of all flowers. On the various dates, the flowering intensities of the orchard were 1.13 %, 3.94 %, 19.54 %, 37.34 %, 57.57 %, 72.18 %, 82.26 %, 81.59 %, 75.74 %, 77.04 %, 76.59 %, 47.83 %, and 12.18 %. In addition, the orchard was at its first flowering stage on 3 April, its middle flowering stage on 5 April, its full flowering stage on 7 April, and its last flowering stage on 15 April.

The apple flower monitoring method proposed in this study is applicable to orchard environments in the open world. Based on the detection of four stages of tiny flowers in complete fruit tree images, the quantitative analysis of data and the assessment of blossom intensity were realized, and then the flower information monitoring was realized. It is important to note that the existence of diverse viewing angles, illumination fluctuations, occlusions, uncertain stances, low pixel ratio, complicated backdrops, etc., makes it challenging for models trained on the source dataset to achieve high performance. This method establishes the foundation for the proper use of IoT technology for the remote monitoring of flowering information in modern orchards.

**Author Contributions:** Conceptualization, X.Z. (Xinzhu Zhou); methodology, X.Z. (Xinzhu Zhou) and G.S.; software, X.Z. (Xinzhu Zhou) and X.Z. (Xiaolei Zhang); validation, X.Z. (Xinzhu Zhou) and N.X.; formal analysis, X.Z. (Xinzhu Zhou), Y.Y. and J.C.; investigation, X.Z. (Xinzhu Zhou), J.C. and Y.H.; writing—original draft preparation, X.Z. (Xinzhu Zhou), G.S. and N.X.; writing—review and editing, X.Z. (Xinzhu Zhou), Y.Y. and J.C.; project administration, G.S.; funding acquisition, G.S. All authors have read and agreed to the published version of the manuscript.

**Funding:** This work was supported by the key R&D Program of Jiangsu Province (No. BE2022363), High-end Foreign Experts Recruitment Plan of China (No. G2021145009L), Jiangsu agricultural science and technology Innovation Fund (No. CX(22)3097), and Jiangsu agricultural science and technology Innovation Fund (No. CX(21)2006).

**Institutional Review Board Statement:** Not applicable.

**Data Availability Statement:** The data supporting this study's findings are available from the corresponding author upon reasonable request.

**Acknowledgments:** The author would like to thank the editors and reviewers for their comments on improving the quality of this work and MDPI for their English language revisions.

**Conflicts of Interest:** The authors declare no conflict of interest.

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
