# Peer review of "A Method of Modern Standardized Apple Orchard Flowering Monitoring Based on S-YOLO"

_agriculture, doi:10.3390/agriculture13020380_

Round 1

Reviewer 1 Report

The manuscript proposed a Transformer-based open-world apple blossom monitoring method, which achieves high-precision detection of four-stage blossoms, quantity statistics, peak time determination, and blossom intensity analysis by referring to the SAHI algorithm and improving YOLOX. This method lays the foundation for remote monitoring of flowering information in modern orchards by IoT technology.

A few suggestions may improve the quality of the manuscript. The specific suggestions are as follows.

1. The title is "A method ...... based on Transformer", but the manuscript repeatedly highlights the improved YOLO model. It is suggested to change the title to S-YOLO based.

2. The manuscript repeatedly emphasizes "under different weather conditions" but does not analyze the effect of different weather on the experimental results.

3. The critical S-YOLO-s Loss and P-R curves are missing.

4. Some information in Tables 8 and 9 is redundant and should be streamlined.

5. Does the definition of "End open stage" mean the petals fall off entirely or from the beginning? If the latter, how to distinguish between natural phenomena, such as being blown off by the wind, and unnatural phenomena, such as artificial flower thinning?

6. All the best results in the table should be bolded.

7. The image in figure 5 is blurred, so images need to be changed together with other figures.

Reviewer 2 Report

1. In the manuscript, the phrase "under different weather conditions" appears several times, but there is no further analysis of the effect of different weather on the experimental results.

2. What is the difference between mAP and "each type of mAP" in the abstract for the four indicators? The manuscript needs to be clarified.

3. The data in Table 6 are already reflected in Table 7, so it may be redundant to list them separately.

4. What is the reason for the significantly higher YOLOx-l and S-YOLO-l in Table 8 than in Table 9?

5. Some information in Tables 8 and 9 is redundant and should be streamlined.

6. The optimal results in each table are suggested to be bolded, and the incremental results are removed.

Reviewer 3 Report

The manuscript entitled: "A Method of Modern Standardized Apple Orchard Flowering Monitoring Based on Transformer" describes the transformer-based flowering period monitoring approach in an open field in order to better monitor the whole blooming time of modern standardized orchards utilizing IoT technologies.

The manuscript does show two different methods for monitoring flower buds in apple orchard. However, I couldn’t find the traditional way or ways that used for monitoring apple orchard flowering; instead, it was a modified method. Thus, this article must fixed the title and whole parts of it since its missing a control. I have some comments in specific lines:

Line 44: give example of traditional flowering monitoring.

Line 45: I don't really know what has the pruning in a specific day to do in comparison among different flowering monitoring?

Line 46-49: Here it is hard to make a general assessment using a single plant then line 50 and beyond the solution would be using a modern tech method.

Line 54-56: Need a reference here. 

Round 2

Reviewer 3 Report

Dear authors,

Thank you very much for the taking care of the comments and suggestions